# Concentration Polarization Enabled Reactive Coating of Nanofiltration Membranes with Zwitterionic Hydrogel

**DOI:** 10.3390/membranes11030187

**Published:** 2021-03-09

**Authors:** Patrick May, Soraya Laghmari, Mathias Ulbricht

**Affiliations:** Lehrstuhl für Technische Chemie II and Center for Water and Environmental Research (ZWU), Universität Duisburg-Essen, 45141 Essen, Germany; p.may89@posteo.de (P.M.); soraya.laghmari@web.de (S.L.)

**Keywords:** membrane surface modification, anti-fouling, hydrogel, polyzwitterion

## Abstract

In this study, the bottleneck challenge of membrane fouling is addressed via establishing a scalable concentration polarization (CP) enabled and surface-selective hydrogel coating using zwitterionic cross-linkable macromolecules as building blocks. First, a novel methacrylate-based copolymer with sulfobetain and methacrylate side groups was prepared in a simple three-step synthesis. Polymer gelation initiated by a redox initiator system (ammonium persulfate and tetramethylethylenediamine) for radical cross-linking was studied in bulk in order to identify minimum (“critical”) concentrations to obtain a hydrogel. In situ reactive coating of a polyamide nanofiltration membrane was achieved via filtration of a mixture of the reactive compounds, utilizing CP to meet critical gelation conditions solely within the boundary layer. Because the feasibility was studied and demonstrated in dead-end filtration mode, the variable extent of CP was estimated in the frame of the film model, with an iterative calculation using experimental data as input. This allowed to discuss the influence of parameters such as solution composition or filtration rate on the actual polymer concentration and resulting hydrogel formation at the membrane surface. The zwitterionic hydrogel-coated membranes exhibited lower surface charge and higher flux during protein filtration, both compared to pristine membranes. Salt rejection was found to remain unchanged. Results further reveal that the hydrogel coating thickness and consequently the reduction in membrane permeance due to the coating can be tuned by variation of filtration time and polymer feed concentration, illustrating the novel modification method’s promising potential for scale-up to real applications.

## 1. Introduction

With the massive population growth, the exponential technological advancements and on-going globalization and industrialization of the planet, the human species encounters a variety of novel challenges. Among them, severe water scarcity is considered especially precarious. Although water covers around 70% of earth’s surface, less than 1% is easily accessible freshwater. In addition, the global demand for freshwater is expected to increase and, thus, water is estimated to lack in stressed regions already by the year 2025 [1]. In order to sustain life and biodiversity, improve public health, and guarantee economic prosperity as well as political stability, governments and industry nowadays empower research on modern low-cost separation technologies [2,3]. Particularly, membrane separation processes were identified as a key technology, which can address the issue of freshwater scarcity. Inspired by nature, membranes retain undesired substances by size-exclusion or specific interaction mechanisms and, thus, enable desalination ocean water or treatment of contaminated freshwater resources [4]. Moreover, pressure-driven membrane separations are, compared to conventional thermal separation technologies such as distillation, less energy consuming and hence, economically and ecologically more attractive alternatives to increase the water supply [5]. 

However, membranes still suffer from two major limitations: (a) permeability/selectivity trade-off and (b) fouling [6]. While the prior aspect relates to the inverse relationship between water flux and solute rejection, fouling phenomena describe the accumulation of retained organic, inorganic, and biological matter on the membrane surface and/or within permeation channels [7]. The deposition of substances adds an additional hydraulic resistance and results in a flux decline. In order to regain original permeability, membranes need to be treated frequently with intense chemical cleaning or must even be replaced completely. Hence, prevention of fouling is a crucial challenge in membrane application and research [6]. Generally, the deposition of foulants is a multivariable function of feed water characteristics, operational parameters, membrane chemistry, as well as foulant type [8]. More specific, all fouling phenomena are governed by thermodynamic principles and driven by the reduction of Gibbs free energy between foulant, solvent, and membrane [9]. Common strategies to increase fouling resistance of membranes therefore focus on manipulating their surface chemistry, with the purpose to weaken attractive foulant–surface interactions [10,11,12]. Materials, that have empirically proven to reduce fouling phenomena are hydrophilic and overall neutral, and have hydrogen bond acceptor but no donor functionalities [13]. Such physicochemical surface properties are responsible for material’s ability to undergo attractive interactions with water, which forms a protective hydration shell [13]. In addition, maintaining such properties across a broad pH range helps to inhibit non-specific adsorption of proteins as well as adhesion of biological organisms [6,9]. 

In recent years, polyzwitterionic hydrogel structures have gained attention as highly promising fouling-resistant materials, since they combine the non-fouling criteria in a unique way [9,12,14,15,16,17]. Compared to non-charged hydrophilic polymers, polyzwitterions can bind water in a more perturbed structure and, therefore, entropically disfavor its release during adsorption or adhesion of foulants [9]. In addition, polyzwitterions undergo intra- or interchain Coulomb interactions and, consequently, minimize their net charge in self-assembled conformations, which inhibits specific, charge-driven adsorption phenomena [18]. Moreover, high water permeation capacity can be guaranteed when zwitterionic materials are implemented as hydrogel networks and thus offer a crucial advantage in membrane applications. Besides that, such three-dimensional networks have also been shown to reduce the foulant’s attachment to the underlying surface by a size-exclusion mechanism [19]. Thus, some of the recent membrane post-modification methods aim at combining the hydrogel’s and polyzwitterion’s advantages in novel surface coatings [15,20,21,22]. Nevertheless, facile, controllable, and scalable implementation methods for commercial membranes are not very well established [23]. Some techniques utilized photo-initiated radical “grafting-from” polymerizations, whereby zwitterionic monomers form cross-linked hydrogels at the membrane surface. However, photo-initiated reactions cannot be applied to hidden surfaces and can lead to material degradation [20]. Such limitations were addressed by a surface-selective zwitterionic hydrogel modification approach developed by Quilitzsch et al. [24]. Therefore, a macromolecular co-initiator, which possessed surface linker as well as redox-catalyzing functionalities, was first physically adsorbed onto the membrane surface in order to guarantee stable attachment of coating. Reactive coating with a zwitterionic hydrogel was subsequently accomplished by bringing the pre-modified membrane in contact with a reactive solution containing a zwitterionic monomer, a cross-linker monomer, and a redox-initiator. Since the macromolecular co-initiator was not able (due to its molecular weight) to enter the active layer pores of the ultrafiltration membrane, the coating was restricted to outer membrane surfaces. The method was also applied in hollow fiber ultrafiltration membrane modules, demonstrating its scalability potential and advantages by accessing hidden surfaces [24]. 

Freger and coworkers [25] established an alternative approach for membrane surface-selective modification reaction, in which the concentration polarization (CP) phenomenon was utilized with various functional monomers in combination with a redox-initiator system. Generally, CP describes the exponential increase in concentration of retained substances towards the membrane surface during filtration and is a result of convective and diffusive mass transport. Once reactive compounds exceed the necessary (critical) concentration within the boundary layer, a (graft) polymerization reaction is initiated at the membrane surface. The thus established CP-enhanced graft copolymerization to modify the membrane was applicable in dead end as well as in crossflow mode and degree of modification could be adjusted by feed composition (in particular monomer concentration) or operational parameters (flux or crossflow velocity and filtration time) [26]. The feasibility of the transfer to a reactive surface coating of desalination membranes in spiral-wound modules had also been demonstrated [27]. 

Nevertheless, several challenges regarding in situ modification, enabled by the retention of reactive modification agents by the membrane, are connected to the use of monomers (i.e., “small” molecules). The obstacles include poor transferability to porous membranes due to incomplete retention of monomeric compounds, poor control of hydrogel formation, and structure for the random reaction between functional and cross-linker monomers as well as unknown extent of CP. This study therefore aims to address such limitations and to enable the formation of an anti-fouling hydrogel by utilizing CP-enabled reaction of novel cross-linkable zwitterionic polymeric building blocks, instead of reactive monomers. The concept is schematically depicted in Figure 1. Strong hydrogel attachment on negatively charged polyamide (PA) thin-film composite (TFC) membranes is achieved by previous adsorption of a cationic cross-linkable macromolecular surface linker, inspired by the works of Quilitzsch et al. [24] and Lei and Ulbricht [19]. Preliminary results demonstrated the feasibility of CP-enabled polyzwitterionic modification, but without reporting any details about the influence of modification conditions or mechanistic insights [28]. Hence, the extent of CP is additionally estimated via modified film model, with the purpose to gain understanding into the dynamic conditions within the boundary layer in dead-end mode, to enable controllability of crosslinking reaction and to assess the influence of operational parameters on modification degree.

## 2. Materials and Methods

### 2.1. Materials

The monomers 2-(dimethylamino)ethyl methacrylate (DMAEMA, Sigma Aldrich, Taufkirchen, Germany) and 2-hydroxyethyl methacrylate (HEMA, Sigma Aldrich) were stored for 24 h at 5 °C with inhibitor remover for hydroquinone (Sigma Aldrich) before use. Azobisisobutyronitrile (AIBN, Sigma Aldrich), methacryloyl chloride (MAC, Sigma Aldrich), methyl iodide (Sigma Aldrich), ammonium persulfate (APS, Sigma Aldrich), tetramethylethylenediamine (TEMED, Sigma Aldrich), tetrahydrofuran (THF, Fisher Scientific, Schwerte, Germany), n-hexane (Fisher Scientific), ethanol (Fisher Scientific), triethylamine (TEA, TCI Chemicals Deutschland, Eschborn, Germany), sodium hydroxide (NaOH, Sigma Aldrich), potassium chloride (KCl, Sigma Aldrich), sodium sulfate (Na_2_SO_4_, Sigma Aldrich), myoglobin (Myo, Sigma Aldrich), and bovine serum albinum (BSA, Sigma Aldrich) were all used as received. Proteins were dissolved in 0.01 M phosphate-buffered saline (1 g/L) and pH was adjusted to the isoelectric point (IEP) of BSA (pH 4.8) or Myo (pH 7) with 1 M HCl. 1,3-Propane sultone (PS, Sigma Aldrich) was heated with hot water above its melting point (31 °C) before used in liquid form. Commercially available PA TFC flat-sheet membranes NF270 (DuPont, Neu Isenburg, Germany) were cut in circular shape (diameter 48 mm) before used in dead-end mode filtration cells.

### 2.2. Polymer Synthesis

Synthesis pathways of cationic surface linker and zwitterionic cross-linkable polymer are depicted in Figure 2.

Free radical copolymerization was performed to obtain poly(dimethylaminoethyl methacrylate-co-2-hydroxyethyl methacrylate) (P(DMAEMA-co-HEMA)). For this purpose, the monomers HEMA (20 mol% of total monomer portion) and DMAEMA (80 mol%) were dissolved in THF in a 100 mL two-neck round bottom flask, so that the monomer concentration was 2.5 mol/L. Subsequently, the mixture was degassed for 15 min with argon gas, before AIBN (0.2 wt.%) was added to the solution. The reaction was performed under reflux at 65 °C for 24 h. Afterwards, the viscous solution was allowed to cool down and the copolymer was precipitated in cold n-hexane. Finally, to remove solvents, the polymer was stored under vacuum at 40 °C for 48 h. 

Crosslinkable methacrylate functionalities were introduced by reaction of hydroxyl side groups with MAC. First, 3.0 g of P(DMAEMA-co-HEMA) were dissolved in 75 mL THF in a 250 mL two-neck round bottom flask. The solution was cooled down to 0 °C and triethylamine (1.3-fold excess compared to HEMA units) was added. Subsequently, MAC (10-fold excess compared to HEMA units) was dissolved in 50 mL THF and added dropwise to the solution within one hour. The mixture turned immediately white and after all MAC had been added, reaction was allowed to proceed for 48 h at room temperature. Subsequently, water was added to hydrolyze unreacted MAC. Furthermore, THF was removed via rotary evaporation and the copolymer was precipitated by adjusting the pH to approximately 10 with NaOH (1 M). Thus, obtained poly(dimethylaminoethyl methacrylate-co-2-methacryloylhydroxyethyl methacrylate) (P(DMAEMA-co-MAHEMA)) was dried under vacuum at 40 °C for 24 h.

Zwitterionic groups were obtained by quaternization of dimethylamino groups with 1,3-propane sultone. For that, 2.0 g of P(DMAEMA-co-MAHEMA) were dissolved in 50 mL THF within a 100 mL two-neck round bottom flask. Subsequently, PS (1.5-fold excess compared to DMAEMA units) was added to the solution and the reaction proceeded under reflux at 48 °C for 24 h. The polymer swelled overnight and it was re-dissolved by adding water. Removal of THF by rotary evaporation was followed by removal of PS via dialysis against water (MW cut-off 12–14 kDa) for three days. The dialysate was replaced with fresh water every day. Finally, water was removed via freeze drying to yield zwitterionic poly(sulfobetaine methacrylate-co-2-methacryloylhydroxyethyl methacrylate) (P(SBMA-co-MAHEMA)) was determined.

Analogous to zwitterionic functionalization, cationic macromolecular surface linker was obtained by quaternization of P(DMAEMA-co-MAHEMA) with methyl iodide, instead of PS. For this purpose, 2.0 g of P(DMAEMA-co-MAHEMA) were dissolved in 50 mL THF and added to a 100 mL one-neck round bottom flask. Afterwards, methyl iodide (1.3-fold excess compared to DMAEMA units) was dissolved in 10 mL THF and added dropwise to the copolymer solution within 20 min. The mixture immediately became turbid and the reaction was terminated after 3 h by adding water. Due to its toxicity, methyl iodide was removed with the aid of a cooling trap and, subsequently, THF was evaporated via rotary evaporator. Water was finally removed by freeze drying to yield the copolymer poly(trimethylammoniumethyl methacrylate-co-2-methacryloylhydroxyethyl methacrylate) (P(TMAEMA-co-MAHEMA)).

The molecular weight (MW) of P(DMAEMA-co-HEMA) was determined by using size-exclusion chromatography (SEC) and all copolymer compositions were investigated via ^1^H NMR spectroscopy (see Section 2.3).

### 2.3. Polymer Characterization

#### 2.3.1. Copolymer Composition 

Composition of all copolymers was analyzed via ^1^H NMR spectroscopy using a Bruker DMX300 instrument (Billerica, MA, USA) and D_2_O as solvent. The fraction of zwitterionic and crosslinkable groups in P(SBMA-co-MAHEMA) was determined by the ratio of signals belonging to methacrylate or sulfobetaine functionalities. The spectra of P(SBMA-co-MAHEMA) and the formula for calculation of copolymer composition are shown in Appendix A. 

#### 2.3.2. Molecular Weight 

Molecular weight (MW) and polydispersity index (PDI) of P(DMAEMA-co-HEMA) were analyzed by SEC using a HPLC system comprising a PSSGram column (Polymer Standard Services, PSS, Mainz, Germany). For that purpose, polymer was dissolved in the eluent, i.e., dimethylacetamide containing LiBr (c = 0.01 mol/L), so that concentration was 4 g/L. Eluent flow rate was adjusted via PU-2080 Plus pump (Jasco, Pfungstadt, Germany) to 1 mL/min and polymer concentration at different elution times was determined via a dual detector for viscosity and refractive index (ETA 2020, Dr. Bures GmbH, Berlin, Germany). Samples were analyzed via relative calibration with poly(methyl methacrylate) (PMMA) as standard (obtained from PSS) as well as universal calibration method. To obtain MW of P(SBMA-co-MAHEMA), that is only soluble in water, the experimentally determined value for P(DMAEMA-co-HEMA) was multiplied by a factor considering the side group’s molecular weights (see Appendix A).

#### 2.3.3. Hydrodynamic Size of Copolymer in Solution

Hydrodynamic radius measurements for P(SBMA-co-MAHEMA) were performed via dynamic light scattering. Polymer was dissolved in pure water (0.1 g/L) or KCl solutions (1–5 mM) and analyzed using the Zetasizer Nano ZS (Malvern Instruments, Malvern, UK) with a capillary cell (DTS1070). HeNe laser (4 mW, λ = 633 nm) was used for excitation and scattered light was detected at an angle of 173°. The refractive index of P(SBMA-co-MAMMA) was assumed to be identical to that of PMMA (1.49) and was used for calculation.

#### 2.3.4. Viscosity and Overlap Concentration of Copolymer Solutions

The viscosity of solutions containing P(SBMA-co-MAHEMA) in various concentrations (0–10 wt.% in water) was analyzed using a viscosimeter Physica MCR301 (Anton Paar, Graz, Austria) with a cone-plate geometry in rotation mode. The polymer solution was first homogenized, before 200 µL were placed onto the plate and the gap was adjusted to 0.1 µm. Measurements were performed with varying shear rate (0.01–800 1/s) and at constant temperature of 25 °C. The measurement of viscosity in dependence of increasing concentration allowed to determine the overlap concentration, which is attained when the slope of the plot increases. According to De Gennes [29], above overlap concentration, polymer coils are in direct contact with each other, which is a precondition for gelation of macromolecular solutions. 

### 2.4. Rheological Investigation of Hydrogel Formation in Free Bulk

The hydrogel formation from P(SBMA-co-MAHEMA) building blocks was monitored by in situ rheological measurements using the viscosimeter Physica MCR301 (Anton Paar) with plane-cone geometry in oscillating mode. The crosslinking reaction of polymer solution with different concentrations (4, 5, 7.5, and 10 wt.%) was initiated by adding the redox initiator system APS and TEMED at room temperature. Hereby, the amount of APS was adjusted, so that the mole ratio between APS and methacrylate groups present in P(SBMA-co-MAHEMA) was 1:2, 1:5, or 1:10. However, the ratio of APS to TEMED was kept constant (1:8) in all gelation experiments. The conditions for bulk gelation are summarized in Appendix A. After quick homogenization of the reactive mixture, 400 µL were placed onto the lower plate and the gap between the plates was adjusted to 0.1 mm. Subsequently, the gelation kinetics were investigated by measuring storage modulus (G″) and loss modulus (G′). Gelation point is defined as the time, when storage equals loss modulus. Hydrogel formation could proceed for 120 min and was considered finished, when no further increase in storage modulus was observable. In addition, mechanical hydrogel properties were characterized by the damping factor, which is defined as loss divided by storage modulus at the end of crosslinking reaction, see Equation (1):(1)$$\tan\delta =\frac{G\prime \prime }{G^{\prime }}.$$

### 2.5. Concentration Polarization-Enabled Reactive Coating of NF270 Membranes

#### 2.5.1. Membrane Performance Characterization

Before use for modification or initial testing, NF270 membranes were washed in EtOH-water mixture (1:1) for 1 h, followed by rinsing with pure water overnight. In order to characterize permeability and salt rejection, membrane samples were placed into a stainless-steel dead-end cell. Compaction of membranes was performed at a constant pressure of approximately 15 bar until water flux was constant. Water permeability (P_0_) was determined for all samples used for modifications, measured 3 times for each 3 min and calculated by Equation (2):(2)$$P_{o}=\frac{m_{w}}{\rho \cdot A\cdot t\cdot \Delta p},$$
with m_w_ as mass of permeated water, ρ as density of water at room temperature (0.998 g/mL), A as effective membrane surface area (9.62 cm^2^), t as permeation time and p as transmembrane pressure difference.

Rejection of membranes for salts (NaCl or NaSO_4_; 2 g/L) as well as APS (1 g/L) or TEMED (1 g/L) in water were also measured for selected membrane samples. For this purpose, filtrations of solutions were performed until permeate volume was about 10 mL (100 mL initial feed volume). Salt rejection was determined by measuring conductivity of feed and permeate solution using the conductometer Lab 960 (Schott Instruments) along with a calibration and using Equation (3):(3)$$R\left[\%\right]=\frac{C_{\mathrm{feed}}-C_{\mathrm{permeate}}}{C_{\mathrm{feed}}}\times 100$$
where R is rejection, and C_feed_ and C_permeate_ are concentrations of feed or permeate, respectively.

The rejection of organic substances was estimated via total organic carbon (TOC) measurements (TOC Autosampler ASI-V, Shimadzu) along with a calibration and also using Equation (3).

#### 2.5.2. CP-Enabled Membrane Modification

After the water permeability of the membrane was characterized (cf. Section 2.5.1), surface modification was performed. First, the negatively charged PA barrier layer of the NF 270 membrane was pre-modified by adsorption of cationic surface linker. This was achieved by dissolving cationic surface linker in water (1 g/L) and adding 25 mL into the cell. The adsorption was allowed to take place for 1 h at a stirring rate of 300 rpm without additional pressure. Thereafter, the cell was emptied and the membrane was washed five times with pure water and one time with NaCl solution (2 g/L).

For membrane modification, P(SBMA-co-MAHEMA) was first dissolved in 100 mL water, before redox-initiator was added to the solution. The reactive mixture was added into the dead-end cell and pressure (12–15 bar) was applied immediately, so that the initial flux was adjusted between 18 to 22 L/hm^2^. The modification procedure was monitored by measuring permeate flux every 3 min. The modification was investigated in dependence of polymer and initiator concentration as well as filtration time, see Table 1.

The modification was terminated by releasing the pressure and washing the membrane 10 times with pure water and 2 times with NaCl solution (2 g/L). Afterwards, modified membrane water permeability (P_mod_) was measured as described in Section 2.5.1 and maintained permeability was obtained with Equation (4): (4)$$\mathrm{Maintained}\;\mathrm{permeability}\;\left[\%\right]=\frac{P_{\mathrm{mod}}}{P_{0}}\times 100$$

An example for a typical membrane modification and the corresponding flux during in situ hydrogel formation is given in Figure 3. Membrane properties were further investigated by salt rejection (cf. Section 2.5.1) or fouling experiments (cf. Section 2.5.3), other samples were washed for 24 h with water and subsequently freeze dried, in order to analyze membranes via zeta potential measurements or scanning electron microscopy (Section 2.5.3). 

#### 2.5.3. Membrane Characterization 

**Zeta potential.** Surface charge of pristine, premodified, and hydrogel modified membranes was investigated by performing zeta potential measurements using SurPASS electrokinetic analyzer (Anton Paar). First, membranes were immersed for 20 min in KCl solution (1 mM), before they were fixed in the measuring cell with a gap of 100 ± 5 µm. The pH was first set to 2.5 and after each measurement of streaming potential, the pH was stepwise increased (by about 1 pH unit) by automatic titration with KOH (0.1 mM) and streaming potential was determined again. Streaming potential was measured five times at each pH step and the values were averaged. Zeta potential was calculated using the Helmholtz–Smoluchowski equation. Based on extensive experience with the system, the maximum error for repeated analysis of the same sample using this procedure is ±5 mV.

**Scanning electron microscopy.** Scanning electron microscopy analysis was performed to determine hydrogel layer thicknesses using the Quanta 400 FED microscope (FEI). Samples were first frozen by immersion in liquid nitrogen, before cut and finally sputtered with an Au/Pd alloy (5:1).

**Membrane fouling.** Protein filtration experiments were carried out to investigate fouling propensity of pristine and hydrogel modified membranes. For this purpose, 100 mL of protein solution (BSA or Myo with pH 4.8 or 7.0, respectively, in 0.01 M phosphate-buffered saline) were filled into the dead-end cell, for the hydrogel-coated membranes immediately after modification and washing. The solutions were filtered with an initial flux of approximately 40 L/hm^2^. The flux course was monitored gravimetrically until 50 mL of permeate was collected. The retentate solution was discarded and the membrane was washed 10 times with water and 2 times with NaCl solution (2 g/L). Water permeability was measured again, in order to determine reversible fouling recovery (RFR), see Equation (5):(5)$$\mathrm{RFR}\;\left[\%\right]=\frac{P_{F}}{P_{0}}\times 100$$
with P_F_ as permeability after fouling filtration and washing.

**Hydraulic resistance of the hydrogel.** By utilizing a resistance-in-a-series model (Equations (6) and (7)) and measured membrane permeabilities, hydrogel’s and polyamide layer’s hydraulic resistances were calculated [30]:(6)$$R_{\mathrm{tot}}=R_{\mathrm{mem}}+R_{\mathrm{gel}}$$
(7)$$R_{\mathrm{tot}}=\;\frac{\Delta p}{J_{\mathrm{tot}}}=\frac{\Delta p}{J_{\mathrm{mem}}}+\frac{\Delta p}{J_{\mathrm{gel}}}$$
with R_mem_ as resistance of pristine membrane, R_gel_ as resistance of the hydrogel, R_tot_ as resistance of the hydrogel-coated membrane, Δp as transmembrane pressure, and J as water flux (indices “tot”, “mem” or “gel” for “total”, “membrane” or “hydrogel”, respectively). In addition, thickness independent specific resistance of the hydrogel, R_spec_, can be obtained by using experimentally determined hydrogel layer thicknesses and Equation (8): (8)$$R_{\mathrm{spec}}=\;\frac{R_{\mathrm{gel}}}{\Delta l}$$
with Δl as hydrogel layer thickness obtained from scanning electron microscopy.

### 2.6. Estimation of Concentration Polarization 

#### 2.6.1. Approach 

Extent of concentration polarization was calculated in order to evaluate in situ hydrogel formation at the membrane surface (cf. Figure 1) and to compare boundary layer and bulk gelation conditions. According to the film model, CP in the steady state of filtration is a consequence of convective solute flux towards, diffusive solute flux away from, and solute permeation flux through the membrane, described by Equation (9) [31]: (9)$$J_{v}c_{p}=J_{v}c_{f}-D_{i}\frac{\mathrm{dc}}{\mathrm{dx}}$$
with J_v_ as the permeate flux, D_i_ as diffusion coefficient of solute i, c_p_ as permeate concentration of solute i, c_f_ as feed concentration of solute i, x as the distance from membrane surface, and c as variable concentration of solute i. 

By integrating Equation (9) over the boundary layer thickness *δ* CP modulus (CPM) can be calculated by Equation (10): (10)$$\mathrm{CPM}=\frac{c_{m}-c_{p}}{c_{f}-c_{p}}=e^{\frac{J\delta }{D}}$$
with c_m_ as concentration of solute i at the membrane surface. 

Only the zwitterionic cross-linkable copolymer, as the component of key importance for the formation of a hydrogel, was considered here. However, CPM and c_m_ for the copolymer cannot be measured directly. In addition, no steady-state will be achieved, since modification is performed in dead-end mode. Therefore, in this work, experimental flux data measured during modification (cf. Figure 3) under specific conditions (cf. Table 1) was used to calculate convective copolymer mass transport towards and diffusive copolymer flux away from the membrane, iteratively for subsequent short time intervals (1 s). The following assumptions were made:(i)concentration polarization increases linearly within the boundary layer δ;(ii)boundary layer thickness δ is constant during entire modification procedure;(iii)back diffusion is dependent on macromolecule size and solution viscosity.

In order to calculate extent of CP and copolymer concentration at the membrane surface c_m_ as function of filtration/reaction time t, the following experimental data were used as input (n is number of time intervals, starting with n = 1 for the 1st second of filtration): (i)boundary layer thickness δ, which was estimated during a separate experiment (see Section 2.6.2), along with A as effective membrane area yielding the volume of the boundary layer;(ii)flux J(t_n_) and initial feed concentration c_f_(t_0_), which allow to calculate the permeate volume in the time interval V(t_n_) and thus also the convectively transported mass towards membrane surface m_C_(t_n_) and subsequently also c_f_(t_n_);(iii)single macromolecule radius r_polymer_, which was obtained by DLS measurement (see Section 2.3.3), required to calculate the diffusion coefficient of the polymer and thus diffusive mass transport back toward feed m_D_(t_n_);(iv)average viscosity of copolymer solution in boundary layer η(c_polymer_), using a relationship derived from rheological measurements (see Section 2.3.4) and calculated values for c_f_(t_n_) and c_m_(t_n_), also required to calculate m_D_(t_n_).

The concentrations at the membrane surface c_m_(t_n_) and within the feed c_f_(t_n_) for every subsequent time interval n can be calculated with Equations (11) to (13). Their mathematical derivation is presented in Appendix A:(11)$$c_{m}\left(t_{n}\right)=2*\left(\frac{\left.m(t_{n}\right)}{A*\delta }+{\;c}_{f}\left(t_{n}\right)\right)$$
(12)$$c_{m}\left(t_{n+1}\right)=2*\left(\frac{\left.m(t_{n+1}\right)}{A*\delta }+c_{f}\left(t_{n+1}\right)\right)$$
(13)$$c_{f}\left(t_{n+1}\right)=c_{f}\left(t_{n}\right)+\frac{m_{D}}{V\left(t_{n+1}\right)}$$

#### 2.6.2. Estimation of Boundary Layer Thickness

Boundary layer thickness was estimated by rejection measurements of a semi-permeable solute, using the relation between mass transfer coefficient and rejection (Equation (14)) developed by Koyuncu and Topacik [32]. Since the cross-linkable polymer is completely rejected by the NF270 membrane, the monomer HEMA was used as test solute:(14)$$\ln\left[\frac{1-R_{\mathrm{obs}}}{R_{\mathrm{obs}}}\right]=\ln\left[\frac{1-R_{0}}{R_{0}}\right]+\frac{J}{k}$$

Opong and Zydney [33] demonstrated, that the mass transfer coefficient in stirred filtration cells is dependent on stirring rate and can be calculated by Equation (15):k_d_ = k′_d_ ω^0.567^(15)
with ω as stirring velocity and k’_d_ as cell-specific solute dependent mass transfer coefficient. For no stirring, rate ω was assumed to be 10.6 rpm. 

By combining Equations (14) and (15), R and k’_d_ can be derived from the plot of ln(R/1-R) vs. J_v_ω^−0.567^ as intercept and slope, respectively, and boundary layer thickness can be calculated with Equation (16):(16)$$\delta =\frac{D_{\mathrm{HEMA}}}{k_{d}}$$
with D_HEMA_ as the diffusion coefficient of HEMA in water (5.9 × 10^−10^ m^2^/s) [25]. 

Filtration experiments with HEMA solutions in water were performed at different stirring rates (0–1000 rpm) and constant pressure (15 bar). Feed concentration of HEMA was 0.0009 wt.%. HEMA concentrations were measured using TOC analysis, and rejection was calculated using Equation (4) (cf. Section 2.5.1); 3 experiments were performed for each stirring rate.

## 3. Results and Discussion

### 3.1. Polymer Synthesis and Characterization

The strategy for synthesis of the cross-linkable polyzwitterionic building blocks was adapted from previous own work; the copolymer P(SBMA-co-HEMA) had been established as one of various polyzwitterions that can be cross-linked via “click” reactions with commercial low-molecular weight reagents, specifically with a bis-epoxide for the nucleophilic hydroxyl side groups in P(SBMA-co-HEMA) [22]. In the current work, a straightforward acylation of the hydroxyl groups of the HEMA segments was used to introduce reactive methacrylate units. In literature, it had been shown that such functionalization of PHEMA leads to materials that can form hydrogels upon addition of radical generators or dithiol cross-linking agents in a thiol-Michael “click” type reaction [34,35,36]. The utilization of this polymer-analogous functionalization for P(DMAEMA-co-HEMA) in a quantitative manner, followed by sulfobetainization of the DMAEMA segments, to yield the target polymer has, to the author’s knowledge, not been reported before by another group; the only related reference is own preliminary work [28]. 

#### 3.1.1. Molecular Weight and Composition of P(SBMA-co-MAHEMA)

Table 2 summarizes results for analysis of chemical composition as well as molecular weight of the zwitterionic, cross-linkable copolymer.

The molecular weight of P(SBMA-co-MAMMA) determined via SEC is 122.2 kDa. Additionally, PDI is approximately 2.6, which is common for free radical copolymerization. Thus, macromolecules are sufficiently large to be completely rejected by dense PA membranes, allowing effective accumulation at membrane surface during filtration. 

The NMR spectra Appendix A show, that not more than 1% of hydroxyl or dimethylamino groups remain unfunctionalized during the polymer-analogous reactions. Sulfobetainization of tertiary amines is reported to maintain a small degree of tertiary amine functionalities [22]. The introduction of cross-linkable methacrylate groups can be incomplete due to possible hydrolysis of MAC, although a 10-fold excess of MAC was used. Nevertheless, a high degree of conversion to zwitterionic or cross-linkable groups is achieved; the resulting copolymer contains mainly the two different target side groups.

Overall, the first step is a simple free radial copolymerization of two commercial monomers leading to a random copolymer. The second and third steps are straightforward polymer-analogous reactions that are “orthogonal” with respect of the reactivity. In the second step, only the hydroxyl (-OH) groups of the HEMA segments will be functionalized; in the third step, only the tertiary amino groups (-N(Me)_3_) of the DMAEMA segments will be functionalized. All these features would make upscaling the synthesis easy and relatively low cost.

#### 3.1.2. Hydrodynamic Size of P(SBMA-co-MAMMA) 

Hydrodynamic size of P(SBMA-co-MAMMA) was analyzed via DLS measurements in order to calculate the diffusion coefficient, necessary for estimation of CP (cf. Section 2.6.1). Figure 4 shows number-based hydrodynamic size distribution of polyzwitterionic polymer in pure water and KCl solutions.

Except for highest salt content, all solutions demonstrate polydisperse particle size distribution. In pure water, the hydrodynamic sizes are in the range of 50–180 nm. Hence, results point to extensive agglomerates, which can be attributed to attractive and well-reported inter-chain Coulomb interactions between zwitterionic macromolecules [37,38,39,40]. Such agglomerations may facilitate gelation in water due to proximity of different chains. However, when salt is added, agglomeration is reduced due to charge shielding, diminishing attractive inter-chain interactions [37,40]. The effect is schematically depicted as insets in Figure 4. In 5 mM KCl solutions, particle sizes in the range of 10–25 nm are observed; this is a reasonable range for unimers, i.e., single macromolecule coils. The peak size (14 nm) is utilized to calculate the diffusion coefficient for estimation of CP (see Section 2.6.1). 

#### 3.1.3. Viscosity of Aqueous Solutions Containing P(SBMA-co-MAMMA)

The viscosity of polymer solutions was studied in dependence of concentration. The results are depicted in Figure 5 and demonstrate a linear increase in viscosity with increasing polymer content due to more frequent polymer–solvent and polymer–polymer interactions. Specifically, two different linear viscosity-concentration relationships can be observed. The diluted polymer solution range is observed between 0 and 4 wt.%. Within that concentration regime, macromolecules are not in close enough proximity and, thus, viscosity increase is primarily a result of increasing number of polymer–solvent interactions. Consequently, average distance between different polymer chains is relatively big. In these conditions, cross-linking would require diffusion of chains, which is, compared to radical chain propagation reaction, slow. Hence, methacrylate-functionalized macromolecules cannot form regular chemically cross-linked hydrogel structures upon initiation with radicals. Consequently, intrachain reactions are more likely to occur, resulting in cyclic macromolecules or gel particles. However, above 4 wt.% friction between polymer chains becomes more relevant and such polymer–polymer interactions contribute to a more pronounced viscosity increase with increasing solution concentration; the slope rises from 0.54 to 0.76 mPa s/wt.%. (i.e., by ~40%). Hence, the polymer solution exceeds overlap concentration, which is a precondition for the formation of regular three-dimensional cross-linked networks. 

### 3.2. Bulk Gelation

#### Gelation Kinetics

Kinetics of free bulk gelation were studied in dependence of polymer and redox-initiator concentrations. The measured gelation times are shown in Figure 6. Experiments have been performed only once for each setting, but preliminary experiments with another batch of copolymer having slightly different molecular weight and composition yielded the same trends at slightly different absolute values.

Generally, gelation time decreases exponentially with increasing polymer concentration, and it decreases also with increasing radical initiator content. For example, for the lowest studied polymer concentration, gelation time is reduced from approximately 22 min to less than 5 min, when APS content is increased 5-fold. The influence of redox initiator concentration on gelation time can be noticed for all reactive solutions, independent of polymer concentrations. At higher initiator concentration, a higher polymer radical concentration is created which consequently increases the number of simultaneous cross-linking reactions. 

Furthermore, the drop of gelation time is strongly pronounced when polymer concentration is increased from 5 to 7.5 wt.%, indicating much more effective collisions between macromolecules. More precise, for polymer concentration of 7.5 wt.% and the two higher redox initiator contents, gelation occurs very quickly within 154 or 40 s. Remarkably, further increase in polymer concentration only slightly reduces gelation time (to ~142 or 16 s, respectively). At first sight, such finding is counter-intuitive since higher polymer content should lead to more collisions between macromolecules. Yet, increase in polymer concentration is accompanied by higher viscosity and thus chain mobility is reduced. Consequently, a balance between radical propagation and polymer chain mobility may result in cross-linking conditions in which gelation time becomes less affected by polymer content.

In addition, hydrogels demonstrate similar mechanical properties, e.g., damping factor for all cross-linked networks is approximately 0.1, independent of the gelation conditions. However, storage modulus increases with polymer concentration (see Appendix A). For 5 wt.% polymer solutions, elastic moduli are in the range of several hundred Pa and increase up to several mPa for 7.5 or 10 wt.%, respectively. This tendency may point to differences in hydrogel’s material densities. For higher polymer content, interchain cross-linking is more probable, compared to intrachain reactions. Thus, low concentrated solutions may result in irregular and less dense structures, resulting in reduced storage modulus. Generally, higher storage modulus implies mechanical stability, which is relevant in membrane applications due to high shear stresses and potential damage of hydrogel coating [41]. In contrast, higher cross-linking degree will also impose stronger barrier for water permeation and hence decrease membrane performance [42]. 

It is further relevant to notice, that successful cross-linking could not be observed in 120 min for solutions containing less than 5 wt.% P(SBMA-co-MAHEMA), independent of redox initiator concentration. Although slightly higher, such estimated critical gelation concentration is in good agreement with the concept of overlap concentration (see Section 3.1). On the other hand, it is important for the use of such (potentially) reactive mixture of P(SBMA-co-MAHEMA) and APS/TEMED in CP-enabled membrane coating that they are macroscopically stable at lower concentrations and for times less than 120 min (cf. Section 3.3).

### 3.3. Concentration Polarization-Enabled Hydrogel Coating of a Polyamide Nanofiltration Membrane 

#### 3.3.1. NF270 Membrane Characteristics 

Membrane characteristics were investigated prior to modification and are given in Appendix A. Compared to other NF membranes, the NF270 type shows a high water permeance (11.3 ± 1.0 L/hm^2^ bar), which is attributed to its extremely thin semi-aromatic PA layer (~21 nm) [43]. Furthermore, membranes show significant permeation for monovalent ions (experimental rejection for NaCl and NaSO_4_ is 38% and 90%, respectively) as well as for the components of the redox-initiator system (rejection for TEMED and APS is 37% and 76%, respectively); these separation properties are typical for “loose” NF membranes. 

#### 3.3.2. Influence of Polymer Concentration on Hydrogel Coating during Filtration

Reduction in permeability at the end of the modification procedure (cf. Figure 3) serves as confirmation for successful formation of a hydrogel layer on the membrane surface. Preliminary experiments had revealed that the adsorptive pre-coating of the membrane with the macromolecular surface linker is essential; omitting this step leads to modified membranes with unstable properties, that can be explained by detachment of the hydrogel coating. 

In Figure 7, the impact of polymer feed concentration on maintained permeability is depicted for constant APS and TEMED content and filtration/reaction time (40 min). All solutions are stable and no effect onto membrane permeability is observed after contact of the solution with the membrane without filtration. The chart reveals that permeability decreases linearly (by about 20 to 75%) within a concentration range of 0.003 to 0.01 wt.%. For a polymer feed content above 0.01 wt.%, reduction in water permeability remains constant. Below a feed concentration of 0.003 wt.%, no reduction in permeability can be observed. Thus, at too low feed concentration, the extent of concentration polarization is assumed to be insufficient to initiate cross-linking reactions between macromolecules. In contrast, CP may reach a maximum for higher polymer concentration, presumably because the flux during modification is simultaneously decreased (see Appendix A and estimation of CP below). Hence, results demonstrate the general adjustability of hydrogel resistance (assumed to be related to coating degree) by polymer feed content within a specific range.

Understanding the relationship between feed concentration and coating degree can be enhanced by estimating the polymer concentration at the membrane surface using the experimental flux values in combination with other input values for the modified film model (cf. Section 2.6.1); for results see Figure 8. In general, average concentration during the entire modification procedure as well as peak concentration at the membrane surface are much higher than feed concentration; the estimated values are in a reasonable range and increase with higher polymer feed concentration. For highest feed concentrations (c_f_ = 0.04 or 0.08 wt.%), membrane peak concentrations are around 12 or 14 wt.% (and average concentration 4 or 6 wt.%), respectively, thus confirming the feasibility of hydrogel formation since critical gelation concentration is exceeded. However, gelation on membrane surface for lower feed concentration (c_f_ < 0.04 wt.%) is not supported by model values, because peak concentration is predicted to be lower than 1.5 wt.% (average concentration < 1.2 wt.%). Consequently, degree of CP seems to be under-estimated for low feed concentration, which may be attributed to the assumed linear increase of polymer concentration in the boundary layer (cf. Section 2.6.1). According to film model, concentration will rise exponentially and thus be significantly higher than estimated. 

In contrast, such argumentation would also imply an extremely pronounced CP for high polymer feed concentrations. However, the increase should be limited by macromolecule’s solubility in water, which is not considered in the model. Thus, concentration at the membrane surface may reach a maximum, and this would explain why maintained permeability becomes independent above a certain polymer feed content (c_f_ > 0.01 wt.%; cf. Figure 7). 

#### 3.3.3. Influence of Redox Initiator

The influence of redox-initiator content on maintained permeability with constant polymer and TEMED concentration (0.08 wt.% or no TEMED) is shown in Figure 9.

The data show, that permeability can be completely recovered, when redox-initiator feed content is lower than 0.06 wt.%.; no gelation occurs on the membrane surface. Furthermore, experiments performed without TEMED do not lead to a successful formation of a hydrogel, indicating that reaction rate may be too low in absence of the catalyst and co-initiator [44]. Interestingly, a strong decline in membrane permeability (~80%) is noticeable, once APS feed concentration is increased above 0.06 wt.%. In addition, the imposed resistance is not affected by further increase in APS concentration. Hence, it may be concluded, that a critical threshold concentration of APS is necessary to initiate cross-linking reaction. However, once this concentration is exceeded degree of coating becomes less dependent of redox-initiator content.

#### 3.3.4. Influence of Filtration Time

The influence of filtration time on coating degree was investigated with constant redox-initiator (0.06 wt.%) and polymer feed (0.08 wt.%) concentrations. Impact on maintained permeability is demonstrated in Figure 10. The diagram shows a linear decrease in membrane permeability with longer filtration/reaction time. While modifications for 10 and 15 min reduce permeability only slightly up to 15%, longer modification procedures (20 or 40 min) result in a pronounced flux decline of almost 50 or 70%, respectively. As expected, longer reaction time will result in thicker layers due to on-going incorporation of convectively transported macromolecules. Such hypothesis is based on the assumption that incoming polymer is strongly hindered to penetrate the already formed hydrogel structure and, hence, must react on top of rather than within the hydrogel network, resulting in continuous growth of the hydrogel layer. 

The assumption, that longer filtration time results in thicker gel layers is supported by SEM images (see Figure 11). In detail, for membranes modified within 15 min only, ultrathin coatings with a thickness below 50 nm are realized. However, for 20 min, filtration time gel layer thickness drastically grows to ~1 µm, and thickness further doubles when filtration is allowed to proceed for further 20 min. The results thus indicate that coating degree can be well adjusted by filtration or gelation time. However, the effects do not scale in a linear fashion with filtration time; one qualitative explanation is that flux is declining with filtration time as well Appendix A. 

In order to understand coating’s impact on membrane permeability, hydraulic resistances of the PA barrier layer of the TFC membrane and of the zwitterionic hydrogel layer on top the PA TFC membrane obtained after various filtration times are presented in Figure 12a.

It is not surprising, that hydrogel resistances increase with filtration time and are in the range of 2.0 × 10^11^ to 1.0 × 10^13^ m^−1^, similar to resistance imposed by the PA layer (4.0 × 10^12^ m^−1^) (assuming PA layer thickness is 21 nm as reported) [43]. Nevertheless, results do not take into account the hydrogel coating thickness. 

Hence, specific hydrogel resistances, obtained by normalization to layer thickness, are shown in Figure 12b. Cleary, the PA layer imposes strongest barrier with up to two orders of magnitude higher specific resistances compared to zwitterionic hydrogels. Due to the significant intrinsic resistance attributed to densely cross-linked PA, thin hydrogel layers (<50 nm) do not contribute significantly to total membrane resistance. Specific hydraulic resistances increase with longer modification time. Layers formed within 15 min show 2 times higher values compared to gel networks formed by longer filtration time. This finding points to an asymmetric hydrogel structure with a material density gradient. Hydrogels build earlier during modification and closer to membrane surface may possess higher material density, which contributes stronger to hydraulic resistances. Such effect is well-reported in particle fouling, in which initially deposited foulants results in more densely packed cake layers [45,46]. Similarly, the gelation of zwitterionic polymer next to PA surface may be faster and integrate more macromolecules within the network, due to highest local concentration. On the contrary, later and more membrane-distant formed structures may be highly permeable with bigger mesh sizes, because polymer concentration at the beginning of boundary layer is low and thus polymer density will be reduced. Additionally, regular crosslinking reaction and hydrogel formation may be impeded at initial CP regions by polymer back diffusion.

#### 3.3.5. Other Membrane Properties 

*Zeta potential.* In Figure 13, zeta potential of pristine, pre-modified (adsorption of surface linker) and hydrogel-coated (c_F_ = 0.08 wt.%) membranes are shown in dependence of pH. As expected, unmodified NF270 membranes possess lowest isoelectric point (pH 3–4), which originates from free carboxylic acid groups in the PA layer. Consequently, their overall charge is negative and decreases to −50 mV for pH > 7. When membranes are pre-modified by cationic macromolecular surface linker, neutral surfaces are obtained at approximately pH 7 and surface charge is largely positive (up to 70 mV) under acidic conditions. However, above pH 9, no difference between pristine and pre-modified membrane can be observed. In contrast, cross-linked polyzwitterionic hydrogel surface maintains pronounced electroneutrality over the whole pH range with amplitudes of 15 to −25 mV. The balancing of net charge can be attributed to intra- and interchain Coulomb interactions of sulfonic acid and quaternary ammonium groups within the hydrogel network. 

*Salt rejection*. Hydrogel-coated membranes (with maintained permeability < 50%) demonstrate same salt rejection properties as pristine ones (see Table 3). Mesh sizes are bigger than ion size and, thus, structural properties of hydrogel network do not impact ion rejection. In addition, the zwitterionic hydrogel chemistry greatly reduces net charge and therefore does not impose additional charge-based exclusion mechanism. However, in other reported hydrogel coatings based on PEG, salt rejection was observed to increase [19]. Such gel networks were discussed to improve salt rejection by either sealing high salt permeation defects on polyamide surface or decreasing CP for ions due to lower water flux [31,47]. On the contrary, poly(sulfobetaine methacrylate) coatings were shown to increase salt permeability, overall such materials are described as low selective hydrogels and thus not to enhance desalination properties [48]. 

#### 3.3.6. Membrane Fouling

In order to evaluate the fouling properties, protein filtration experiments were performed with two model proteins at their respective isoelectric point (cf. Section 2.5.3). Overall, no irreversible fouling was observed after filtration of protein solution and thorough membrane washing. The non-fouling properties were expected for zwitterionic neutral hydrogel surfaces. However, also pristine negatively charged NF270 membranes did not experience decrease in permeability. Compared to other commercial NF membranes, the already existing anti-fouling properties are mainly attributed to the smooth PA top layer and its small effective membrane surface area [49,50]. Nevertheless, relative flux reduction during protein filtration is much more pronounced for unmodified PA membrane (~50%; see Figure 14). In contrast, zwitterionic hydrogel coated membranes only suffer a 5 or 10 % decline for the two different proteins. The improved anti-fouling properties of hydrogel-coated membranes generally arise from steric repulsion mechanism, a strong hydration shell and weakened non-specific interactions, which essentially protect the underlying PA surface from contact with proteins during the filtration. In addition, flux reduction is slightly more pronounced for myoglobin, independent of membrane type. On the one hand, the differences in flux reduction between proteins may be a consequence of solution’s pH (BSA 4.8 and Myoglobin 7) and its impact on membrane–protein interactions. This explanation is particularly relevant for pristine membranes, which demonstrate electroneutrality only at acidic pH and, hence, BSA adsorption on membrane is weaker than for myoglobin. However, since zwitterionic hydrogel coatings are uncharged independent of pH, the size of proteins (hydrodynamic diameter d_H_) plays a primary role (BSA MW = 66 kDa or d_H_ = 7.2 nm vs. Myo MW = 16.7 kDa or d_H_ = 4.4 nm). More precisely, diffusion of myoglobin through cross-linked hydrogel is more probable due to its smaller hydrodynamic size. Thus, myoglobin may be able to interact with membrane surface, while BSA is prevented from contact by being sieved within the gel network [19]. 

## 4. Conclusions

This work successfully demonstrates the possibility of surface-selective, CP-enabled cross-linking of tailor-made zwitterionic macromolecules via redox-initiated radical formation on PA TFC nanofiltration membranes. The coating degree (layer thickness) can be adjusted by reaction/filtration time as well as with polymer feed concentration. Thus, ultra-thin anti-fouling hydrogel layers can be generated on the surface (<50 nm), which only slightly impact water permeability. The experiments further indicate that in order to initiate cross-linking, threshold concentrations of cross-linkable polymer and radical initiator must be exceeded. The established model estimates CP in the right order of magnitude; however, it fails to provide accurate information about boundary layer conditions. Hence, calculated values can only serve as an indicator to explain successful gelation. Future studies will focus on the influence of flux on degree of modification, so that mass transport impact on cross-linking can be evaluated separately and utilized for tuning the coating. Alternatively, the cross-linking of the zwitterionic copolymer can be achieved by utilizing “click” chemistry (e.g., using a thiol as cross-linker, according to the Michael addition mechanism) with its advantages to be a two-component system (instead of a three-component system like in this work). Finally, the transfer of CP-enabled hydrogel formation from cross-linkable macromolecular building blocks will be transferred to the coating of membranes already assembled in modules, in order to expand the potential of this methodology for a wider range of industrially attractive applications.

## Figures and Tables

**Figure 1 membranes-11-00187-f001:**
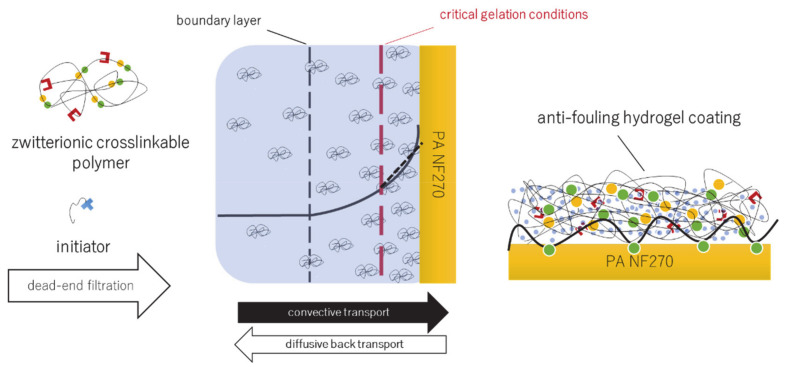
Concentration polarization-enabled modification of nanofiltration membrane surfaces with anti-fouling hydrogel coating using zwitterionic cross-linkable macromolecules as building blocks; when the polymer concentration exceeds a critical concentration, gelation occurs via redox initiated radial cross-linking of C=C double bonds.

**Figure 2 membranes-11-00187-f002:**
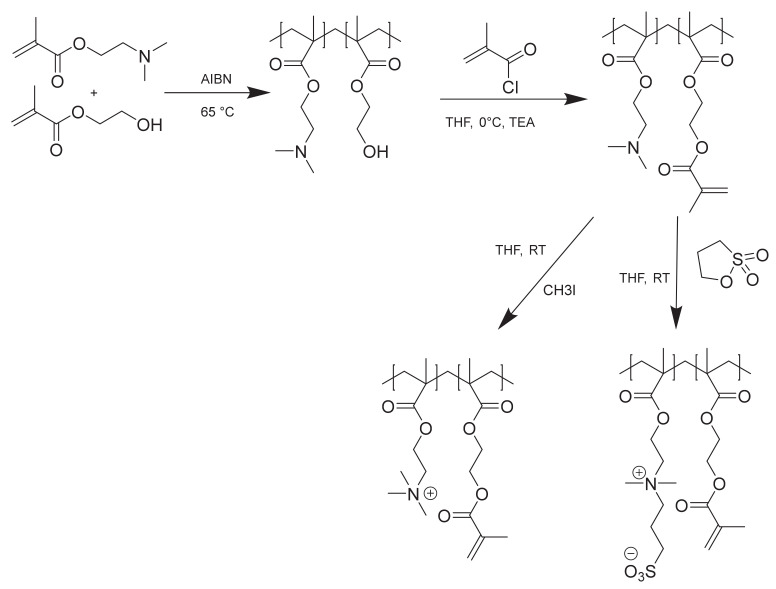
Three-step synthesis of cationic surface linker P(TMAEMA-co-MAHEMA) (below left) and zwitterionic cross-linkable copolymer P(SBMA-co-MAHEMA) (below right).

**Figure 3 membranes-11-00187-f003:**
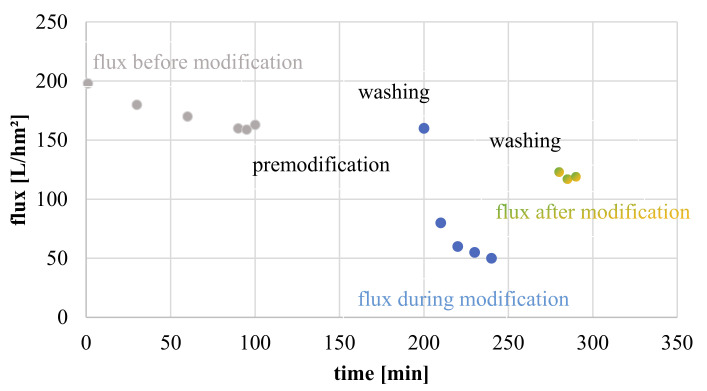
Flux before modification (pure water flux); during modification and after modification (pure water flux).

**Figure 4 membranes-11-00187-f004:**
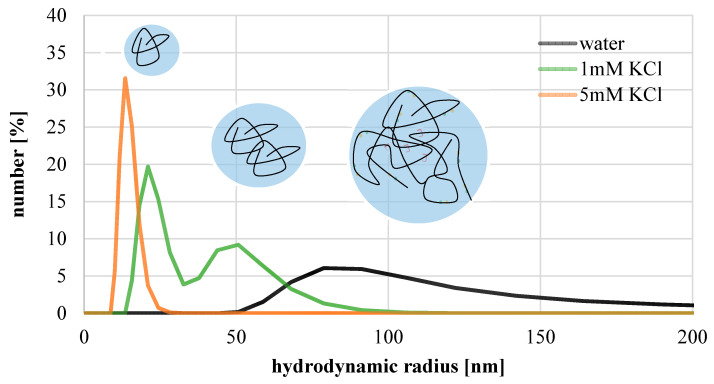
Number-based hydrodynamic diameter of zwitterionic polymer in different solutions.

**Figure 5 membranes-11-00187-f005:**
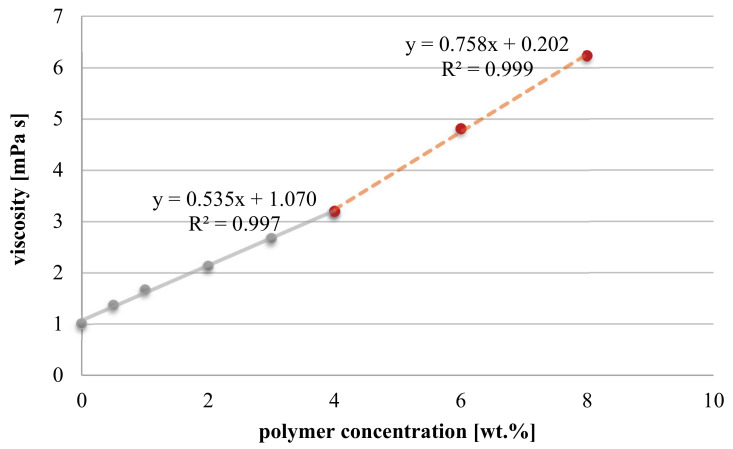
Viscosity of aqueous P(SBMA-co-MAMMA) solution in dependence of polymer concentration (at 25 °C).

**Figure 6 membranes-11-00187-f006:**
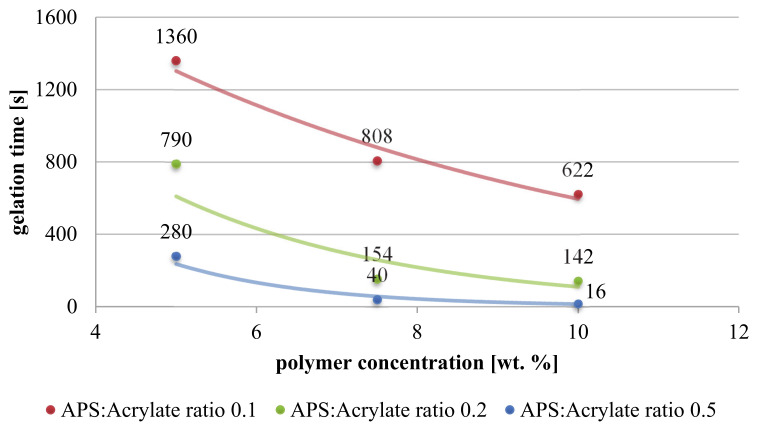
Gelation time in dependence of polymer concentration and APS:Acrylate ratio (APS:TEMED ratio 1:8) at 25 °C.

**Figure 7 membranes-11-00187-f007:**
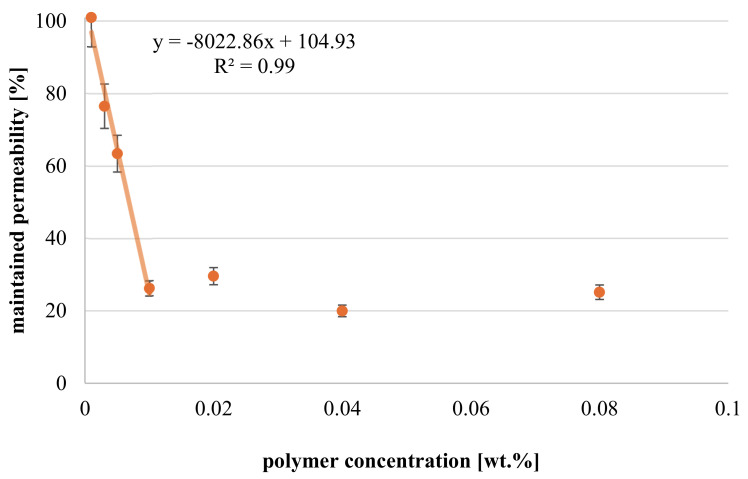
Maintained permeability in dependence of polymer feed concentration (APS 0.06 wt.% and TEMED 0.2 wt.%) used during a modification by filtration (cf. Figure 3).

**Figure 8 membranes-11-00187-f008:**
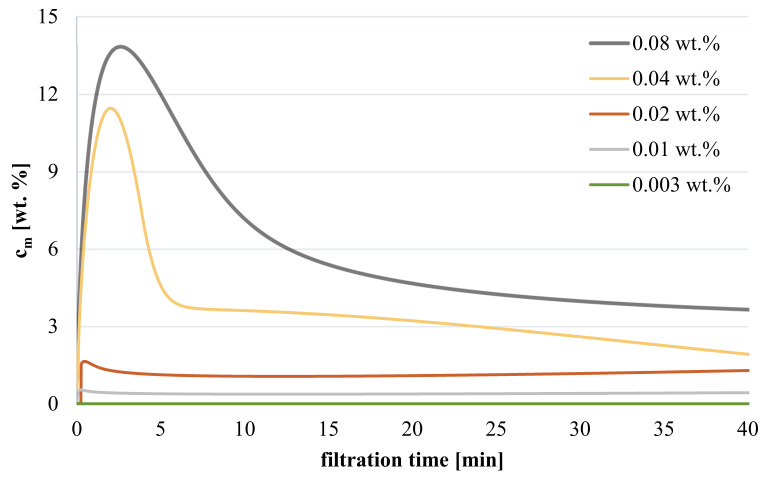
Estimated concentration at the membrane surface in dependence of filtration time for various polymer feed concentrations (cf. Figure 7).

**Figure 9 membranes-11-00187-f009:**
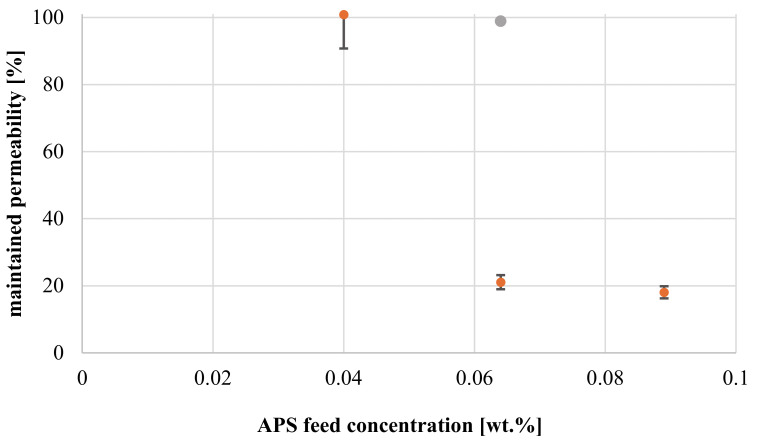
Maintained permeabilities for hydrogel-coated membranes at constant polymer concentration (0.8 wt.%) and constant TEMED concentration (orange: 0.08 wt.%; grey: no TEMED) in dependence of APS feed concentration.

**Figure 10 membranes-11-00187-f010:**
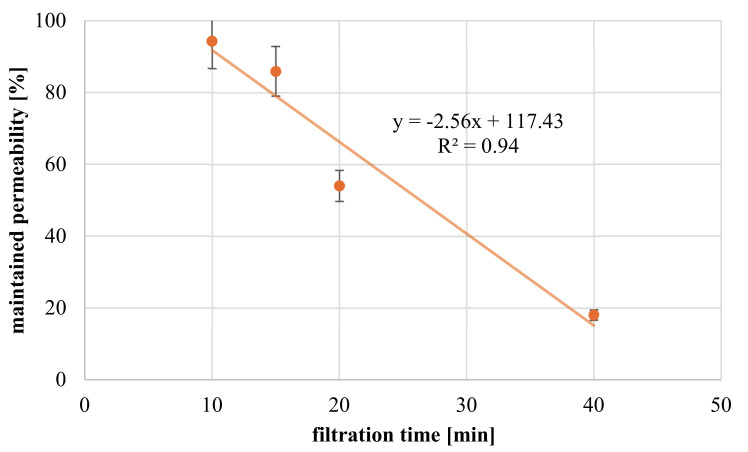
Maintained permeability for hydrogel-coated membranes in dependence of filtration time at constant polymer feed concentration (0.08 wt.%) and constant APS concentration (0.06 wt.%).

**Figure 11 membranes-11-00187-f011:**
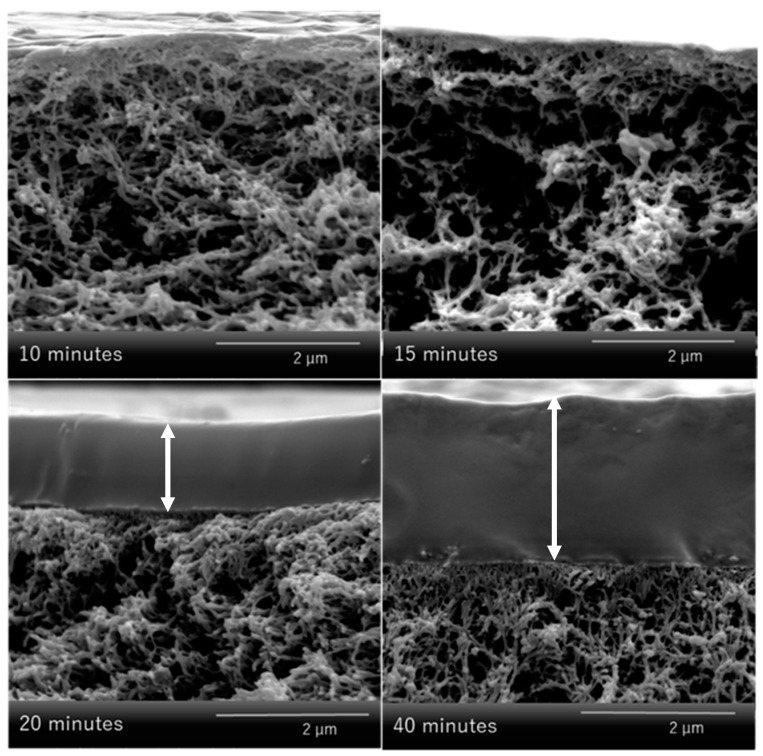
SEM images for cross-sections of hydrogel-coated membranes obtained after different filtration times (cf. Figure 10); white arrows indicate thickness of hydrogel coating.

**Figure 12 membranes-11-00187-f012:**
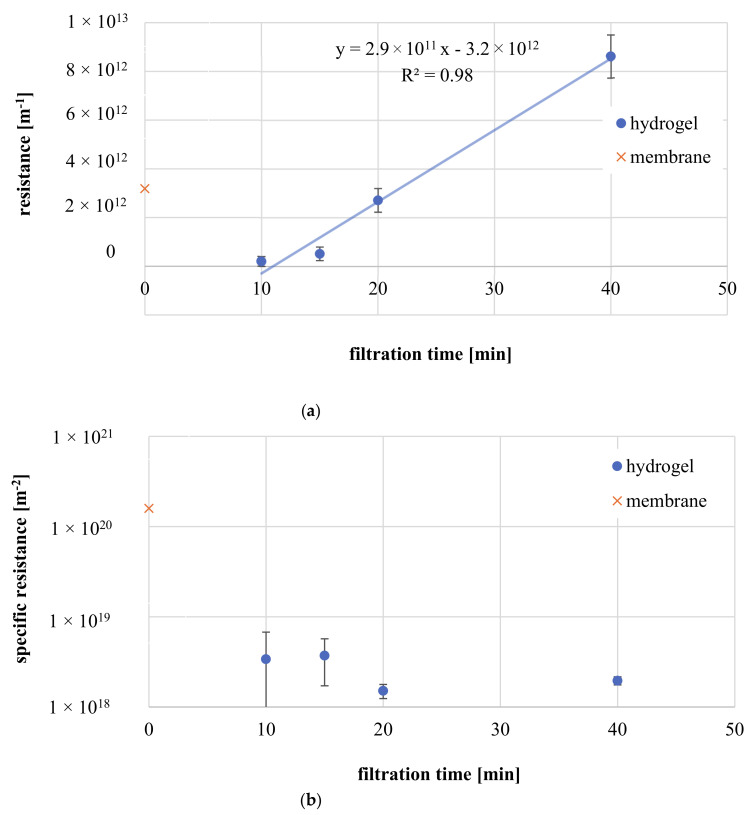
(**a**) Hydrogel resistance for hydrogel-coated membranes obtained after different filtration times, in comparison with the resistance of the PA membrane. (**b**) Specific hydrogel resistance in dependence of filtration time, in comparison with the resistance of the PA membrane (calculated with thickness reported in literature: 21 nm [40]).

**Figure 13 membranes-11-00187-f013:**
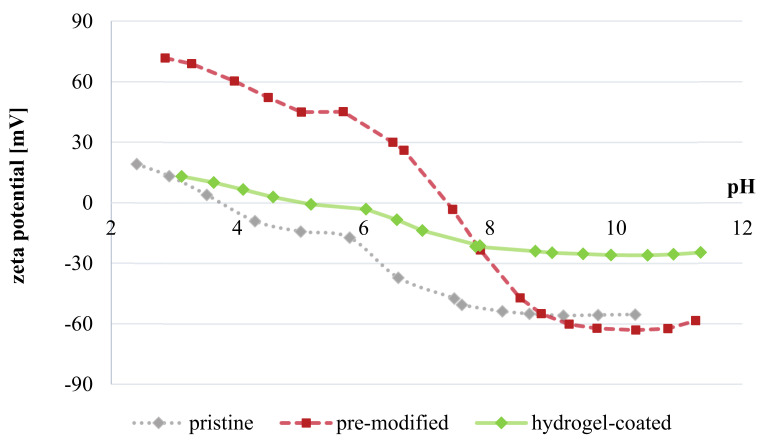
Zeta potential for pristine, pre-modified and hydrogel-coated NF270 membranes (experimental error is < ± 5 mV).

**Figure 14 membranes-11-00187-f014:**
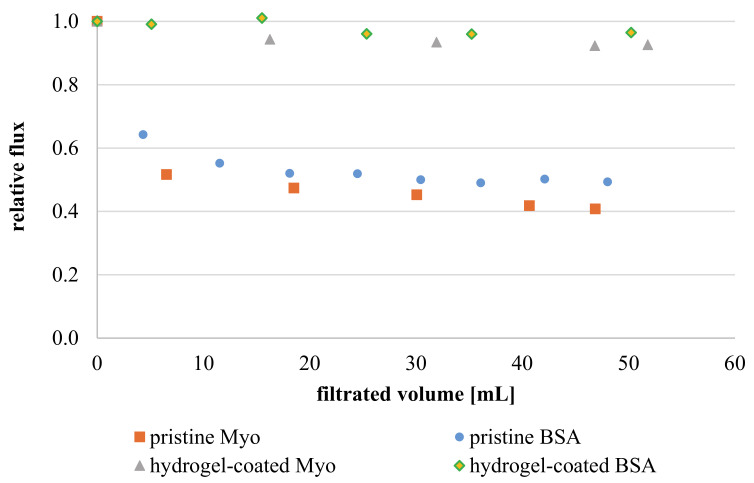
Dynamic fouling experiment: Relative flux courses during filtration of protein solutions (1 g/L, pH 4.8 for BSA and pH 7.0 for Myo) through pristine and hydrogel-coated membranes (results of single experiments for each condition).

**Table 1 membranes-11-00187-t001:** Varied feed parameters at constant TEMED feed concentration (0.06 wt.%) and filtration/reaction time during modification of NF270 membrane. * modification performed without TEMED.

Polymer Feed Concentration [wt.%]	APS Feed Concentration [wt.%]	Reaction Time [min]	No. of Repetitions
0.08	0.06	40	4
0.04	0.06	40	2
0.02	0.06	40	2
0.01	0.06	40	1
0.005	0.06	40	1
0.003	0.06	40	1
0.001	0.06	40	1
0.08	0.09	40	1
0.08	0.04	40	1
0.08	0.06 *	40	2
0.08	0.06	10	2
0.08	0.06	15	2
0.08	0.06	20	2

**Table 2 membranes-11-00187-t002:** Composition and molecular weight of P(SBMA-co-MAMMA).

Methacrylate [%]	Hydroxyl [%]	Sulfobetaine [%]	Dimethylamino [%]	M_n_ (univ.) [kDa]	PDI (univ.)
~20	<1	~79	<1	122.2	2.6

**Table 3 membranes-11-00187-t003:** Comparison of salt rejection between pristine and zwitterionic hydrogel coated membranes.

Characteristics	Hydrogel Coated	Pristine
NaCl rejection [%]	39.3 ± 8.8	37.7 ± 7.8
NaSO_4_ rejection [%]	89.9 ± 7.7	89.7 ± 2.8

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
