# Peer review of "Concentration Polarization Enabled Reactive Coating of Nanofiltration Membranes with Zwitterionic Hydrogel"

_membranes, 2021, doi:10.3390/membranes11030187_

Round 1
Reviewer 1 Report
The manuscript title “Concentration polarization enabled reactive coating of nanofiltration membranes with zwitterionic hydrogel” reported by May et al. the fabrication of surface-selective hydrogel coating using zwitterionic cross-linkable macromolecules as building blocks. The as prepared membrane was examined the different analytical techniques. Then the membrane was performed the salt rejection studies of nanofiltration. I suggested comments below:
- I didn’t see any specific results in abstract selection. the authors remove line 10 “a novel” in abstract part. Should be rewrite in the abstract shown important information and results only presents.
- Could you explain the simple way of polymerization selection?
- 3. polymerization characterization; this selection writing to long, could you short writing is better than the present format.
- 5.1. selection; equation (3) and (4) both same performance, could you rewrite it common equation.
- Should you explain the difference between Figure 7, 8 and Figure 10.
- the authors once again check Figure 9.
- Figure 11; figure caption is not clearly, should be rewrite clearly. Overall the figure caption checks carefully in the manuscript.
- I didn’t see the dense layer of thickness depend on the deposition during time.
- I didn’t see the any specific results.
- Should be comparted pervious reported article.
- The journal format is not fit, still have same problem.
- Should be shown the BSA rejection performance.
- I didn’t see the any stability studies. How to say a wider range of industrially attractive applications of membrane?
- The manuscript have a typo error or mistake.
Reviewer 2 Report
In the manuscript "Concentration polarization enabled reactive coating of nanofiltration membranes with zwitterionic hydrogel," the Authors report the bottleneck challenge of membrane fouling is addressed via establishing a scalable concentration polarization (CP) enabled and surface-selective hydrogel coating using zwitterionic cross-linkable macromolecules as building blocks.
This article is interesting. However, there are some general points in the text which need to addressed before publication:
- Materials and Methods section (page 5 lines 321-327) needs improvement – too short description of the zeta potential measurements. How many measurements for each pH were made? What is its repeatability?
- Figure 6 (page 14), Figure 12 (page 20), Figure 13 (page 21) and Figure 14 (page 23) - What are the error bars of the obtained experimental values? Were presented in these Figures experiments carried out only once?
Round 2
Reviewer 1 Report
The authors have responded the comments.